# Spatio-Temporal Characteristics of PM_2.5_ Concentrations in China Based on Multiple Sources of Data and LUR-GBM during 2016–2021

**DOI:** 10.3390/ijerph19106292

**Published:** 2022-05-22

**Authors:** Hongbin Dai, Guangqiu Huang, Jingjing Wang, Huibin Zeng, Fangyu Zhou

**Affiliations:** 1School of Management, Xi’an University of Architecture and Technology, Xi’an 710055, China; gqhuang@xauat.edu.cn (G.H.); zenghuibin@xauat.edu.cn (H.Z.); 2College of Vocational and Technical Education, Guangxi Science & Technology of Normal University, Laibin 546199, China; 3Chengdu Institute, School of Applied English, Sichuan International Studies University, Chengdu 611844, China; suansuanjunya@gmail.com

**Keywords:** PM_2.5_, remote sensing retrieval, land-use regression, LightGBM, spatial and temporal characteristics

## Abstract

Fine particulate matter (PM_2.5_) has a continuing impact on the environment, climate change and human health. In order to improve the accuracy of PM_2.5_ estimation and obtain a continuous spatial distribution of PM_2.5_ concentration, this paper proposes a LUR-GBM model based on land-use regression (LUR), the Kriging method and LightGBM (light gradient boosting machine). Firstly, this study modelled the spatial distribution of PM_2.5_ in the Chinese region by obtaining PM_2.5_ concentration data from monitoring stations in the Chinese study region and established a PM_2.5_ mass concentration estimation method based on the LUR-GBM model by combining data on land use type, meteorology, topography, vegetation index, population density, traffic and pollution sources. Secondly, the performance of the LUR-GBM model was evaluated by a ten-fold cross-validation method based on samples, stations and time. Finally, the results of the model proposed in this paper are compared with those of the back propagation neural network (BPNN), deep neural network (DNN), random forest (RF), XGBoost and LightGBM models. The results show that the prediction accuracy of the LUR-GBM model is better than other models, with the R^2^ of the model reaching 0.964 (spring), 0.91 (summer), 0.967 (autumn), 0.98 (winter) and 0.976 (average for 2016–2021) for each season and annual average, respectively. It can be seen that the LUR-GBM model has good applicability in simulating the spatial distribution of PM_2.5_ concentrations in China. The spatial distribution of PM_2.5_ concentrations in the Chinese region shows a clear characteristic of high in the east and low in the west, and the spatial distribution is strongly influenced by topographical factors. The seasonal variation in mean concentration values is marked by low summer and high winter values. The results of this study can provide a scientific basis for the prevention and control of regional PM_2.5_ pollution in China and can also provide new ideas for the acquisition of data on the spatial distribution of PM_2.5_ concentrations within cities.

## 1. Introduction

In response to the growing air pollution problem, China has set up large-scale ground-based PM_2.5_ monitoring stations to monitor and warn of heavily polluted weather [1]. PM_2.5_ can largely reduce the body’s immunity and cause respiratory diseases such as asthma and chronic bronchitis, as well as cardiovascular diseases such as heart disease and atherosclerosis, and can increase the risk of cancer [2]. The 2019 Global Burden of Disease Study reports that air pollution is the leading environmental risk factor for global health and the fourth leading risk factor for global mortality, with the disability-adjusted annual loss of life due to PM_2.5_ pollution increasing to 118 million in 2019 and the number of deaths increasing to 4.14 million [3]. Global publicly available PM_2.5_ concentration monitoring data, estimation data and studies on the evolution of national-scale PM_2.5_ concentrations worldwide show that China is one of the countries with high and fast-growing PM_2.5_ concentrations worldwide [4,5,6,7]. A related study estimated the number of deaths caused by PM_2.5_ in 161 major cities in China and showed that PM_2.5_ exposure caused about 652,000 premature deaths in 2015 [8]. Air pollution has become an important environmental problem in China, and accurate prediction of PM_2.5_ concentrations has an important impact on air pollution prevention and sustainable economic development. Aerosol optical depth (AOD) products from satellite remote sensing inversions have been widely used for PM_2.5_ estimation on a global scale [9]. Earlier studies used one-dimensional linear regression models using only AOD as an indicator to estimate PM_2.5_ concentrations or more sophisticated multiple or generalised linear regression models to estimate PM_2.5_ concentrations [10]. Subsequent studies have taken into account surface and meteorological parameters to improve the accuracy of PM_2.5_ estimation [11]. The distribution of PM_2.5_ concentrations is a non-linear process related to a number of factors, with strong temporal and spatial variability [12]. Thus, more sophisticated models have been developed to describe the spatial and temporal variability in the relationship between PM_2.5_ concentrations and AOD, such as geographically weighted regression models [13], mixed-effects models [14] and generalised weighted mixed models [15]. The complex relationship between PM_2.5_, AOD and other indicators is simplified within the model, leaving a large uncertainty in the PM_2.5_ concentration estimates. With the development of computer technology, machine learning (including deep learning) methods are increasingly used in the estimation of PM_2.5_ concentrations due to their powerful non-linear modelling capabilities [16,17]. Such as support vector regression models [18,19], random forest models [20,21], artificial neural network models [22,23], Bayesian methods [24,25], generalised regression neural network models [26,27] and long and short-term memory networks [28], all of which have shown better performance than traditional statistical models in the estimation of PM_2.5_ concentrations. In terms of the selection of influencing factors, these machine learning models used PM_2.5_ information including adjacent temporal and spatial observations [29], land use information [30], vegetation index information [31], nitrogen dioxide (NO_2_) concentration information [32], population density [33] and elevation [34], in addition to AOD and conventional meteorological observation parameters. However, too many hand-designed features are not only time-consuming and labour-intensive, but also too complex for the engineering implementation of the model. In addition, the current model, although effective in reducing the complexity of the objective function, ignores the spatial and temporal variability of PM_2.5_ concentrations. In order to effectively estimate PM_2.5_ concentrations at spatial and temporal scales, a model with better non-linear expression capability and easy engineering is needed. PM_2.5_ concentration data can be obtained through both ground-based monitoring and satellite remote sensing monitoring. The number of ground-based monitoring sites is usually limited and can only reflect local pollutant concentrations at the monitoring sites, which cannot reveal the spatial heterogeneity of PM_2.5_ concentrations within a large study area, which poses a great challenge to the spatial characterisation of PM_2.5_ pollution. The spatio-temporal distribution of PM_2.5_ concentrations has been used to estimate the spatio-temporal distribution of PM_2.5_ [35], but the models are still relatively small and rely heavily on manual feature selection, which does not take full advantage to express highly complex objective functions through deeper and wider network structures. The existing machine learning models do not take into account the spatial and temporal characteristics of PM_2.5_. To this end, there is an urgent need to develop machine learning models that take into account land use information, correlation and spatio-temporal heterogeneity. Accurately revealing the spatial and temporal distribution characteristics of PM_2.5_ concentrations is important for formulating PM_2.5_ pollution prevention, control and management measures.

The contributions of this study are as follows:(1)This study uses an integrated approach combining the LUR model, Kriging method and LightGBM model to improve the daily concentration estimates of PM_2.5_ in the Chinese region from 2016 to 2021. AOD data, latitude and longitude information, meteorological observation elements, land use and road data are used to estimate PM_2.5_ concentrations. Specifically, the accuracy of PM_2.5_ change prediction is improved by stepwise selection of LUR models to identify important predictor variables, and then five machine learning algorithms (BPNN, DNN, RF, XGBoost and LightGBM) are used to build prediction models.(2)The hybrid spatial prediction model proposed in this paper combines the strengths of LUR in identifying the most influential emission predictors. A hybrid spatial prediction model built by identifying the most influential emission predictors combined with LightGBM’s strength in estimating non-linear trends will be more widely effective than traditional machine learning estimation methods. Validated by R^2^, RMSE and MAE metrics, the results show that LUR-GBM performs better.(3)The spring, summer, autumn, winter and 2016–2021 average concentrations are modelled, and the spatial and temporal characteristics of regional PM_2.5_ concentrations in China are analysed.

The rest of the paper is organised as follows. The Section 2 focuses on the data sources used in this study. The Section 3 introduces the methodology and model construction. Section 4 discusses the model results and the spatial and temporal characteristics of PM_2.5_ distribution; Section 5 is the discussion, and Section 6, the conclusions.

## 2. Data Sources

### 2.1. MODIS Remote Sensing Data

The MODIS sensor of NASA is mounted on the Terra and Aqua satellites with multiple channels, featuring multi-spectral, wide coverage and high temporal resolution, which can invert the spatial distribution of AOD data with high accuracy. The MODIS MOD021KM data released from 2016 to 2021 with a spatial resolution of 1 km were used in this work.

### 2.2. PM_2.5_ Site Monitoring Data

PM_2.5_ mass concentration ground-based monitoring data were downloaded from the National Real-Time Urban Air Quality Dissemination Platform, and in this study, daily PM_2.5_ mass concentrations were obtained as daily data for Chinese cities from 2016 to 2021. The distribution of the monitoring stations is shown in Figure 1.

### 2.3. Meteorological Data

The main meteorological data used are planetary boundary layer height (PBLH), relative humidity (RH), air temperature (TEM), surface pressure (SP), wind speed (WIN) and total rainfall (RF). The meteorological data were obtained from the ERA5 data on the European Centre for Medium-Range Weather Forecasts website and were rastered, resampled and cropped using ArcGIS to match the spatial resolution of the AOD data.

### 2.4. Land Use and Road Dataset

This study uses the land-use dataset published by the China Geographic Monitoring Cloud platform. The classification and description of the independent variables are shown in Table 1. Using ArcGIS 10.7 from Esri, Redlands, CA, USA, the land use was classified into six categories, including arable land, forest land, grassland, water, construction land and bare land, after stitching, cropping and reclassification, and considering the area and attributes of each type of land. The road data was obtained from the vector road network of OpenStreetMap, and four categories of highways, trunk roads, primary roads and secondary roads, were extracted within the study area, and the same buffer zones were established with the monitoring station as the centre. The length of each type of road within each buffer zone was obtained as the road factor by the spatial superposition method.

## 3. Methods

### 3.1. LightGBM

The LightGBM algorithm is an improved optimisation algorithm for the gradient boosting decision tree (GBDT) [36]. The model training process was based on a sufficient amount of sample data, and the final output of the model was determined by building multiple decision trees (weak learners) and combining the outputs of the decision tree clusters. The actual training process can be expressed as follows: the decision trees are added in an iterative manner, and when the increase in accuracy due to tree addition is less than a certain threshold, the iteration is stopped and the LightGBM model consisting of $N_{tree}$ decision trees is obtained [37].
(1)$$\varphi (PM_{i})=\sum_{k=1}^{N_{tree}}f_{k}(PM_{i})$$
where $PM_{i}$ is the PM_2.5_ influencing factors; $f_{k}(PM_{i})$ is the kth decision tree.

Heuristic information in LightGBM iteration trees can be used as an important measure of features. Therefore, the tree structure-based metric will directly affect the quality of the subset of candidate features and ultimately determine the experimental effectiveness of the original machine learning algorithm. For any given tree structure, PM_Split represents the total number of times each PM_2.5_ influence factor has been partitioned in the iteration tree. PM_Gain represents the level of importance of each PM_2.5_ impact factor characteristic. They are defined as follows:(2)$$PM\_Split=\sum_{t=1}^{K}Split_{t}\;,PM\_Gain=\sum_{t=1}^{K}Gain_{t}$$
where *K* is the *K* decision trees resulting from *K* rounds of iterations.

### 3.2. LUR Model

LUR is an effective method for modelling PM_2.5_ concentrations because of its high simulation accuracy and comprehensive considerations [38]. In this study, a multivariate regression equation, or LUR model, was constructed for PM_2.5_ concentrations in relation to land-use type, topography, meteorology, road traffic, population density and pollution sources. The basic form of the model usually consists of one dependent variable and two or more independent variables and is calculated in equation [39].
(3)$$y=\alpha_{0}+\alpha_{1}PM(x_{1})+\alpha_{2}PM(x_{2})+\ldots +\alpha_{n}PM(x_{n})+\varepsilon$$
where y is the dependent variable and represents the PM_2.5_ concentration value; $PM(x_{1})$, $PM(x_{2})$, …, $PM(x_{n})$ are the different influencing factors of PM_2.5_; $\alpha_{0}$, $\alpha_{1}$, $\alpha_{2}$, …, $\alpha_{n}$ are the coefficient to be determined; ε is the random variable.

### 3.3. Kriging

The basic principle of Kriging’s method is to estimate data at other unobserved locations in space from data at regularly distributed sample points in space [40].

#### 3.3.1. Regionalised Variables

The study area can be considered as a regionalised variable satisfying Kriging’s interpolation condition R(S), $S_{1}$, $S_{2}$, …, $S_{n}$ are the location of PM_2.5_ ground monitoring stations in the area. $R(S_{1})$, $R(S_{2})$, …, $R(S_{3})$ are the observed value of PM_2.5_ at the corresponding station. For a point $S_{0}$ in the region, the spatial attribute $R_{d}(S_{0})$ can be obtained by interpolation with the Kriging method, and the temporal attribute $R_{t}$ can be expressed in terms of the month in which the point is located [41], which can be expressed as:(4)$$R_{d}(S_{0})=\sum_{i=1}^{n}\omega_{i}R(S_{i}),\;R_{t}=m$$
where $R_{d}(S_{0})$ is the spatial attribute of the given point, $\omega_{i}$ is the Kriging weight, $R(S_{i})$ is the monitoring value of the station around the point and m is the month of the given point.

Kriging satisfies the set of optimal coefficients with the smallest difference between the estimated value $R_{d}(S_{0})$ at the station and the true value $R(S_{0})$, while satisfying the condition of unbiased estimation, as follows:(5)$$\underset{\omega i}{\min}V\mathrm{ar}(R_{d}(S_{0})-R(S_{0})),\;E(R_{d}(S_{0})-R(S_{0}))=0$$

#### 3.3.2. Variance Functions

The variance function is the basis of the kriging interpolation method and is a model function used to describe the spatial relationship between PM_2.5_ ground monitoring stations and between stations and pixels. The variance function for the regionalised variable R(S) can be expressed as the semi-variance $\mu (S_{i},S_{j})$ of the difference between the observations at the monitoring station $S_{i}$ and $S_{j}$ as Equation (6):(6)$$\mu (S_{i},S_{j})=\frac{1}{2}E{\left[R_{d}(S_{i})-R(S_{j})\right]}^{2}$$

#### 3.3.3. Equation Solving

The Kriging equation can be obtained by minimising the variance of the unbiased sum estimate in Equation (7).
(7)$$\sum_{i=1}^{n}\omega_{i}\mu (x_{i},x_{j})-\varphi =\mu (x_{0},x_{i}),\;\sum_{i=1}^{n}\omega_{i}=1$$
where φ is the Lagrangian multiplier factor. Solving the above system of equations yields the Kriging weights $\omega_{i}$ and hence the estimated value $R_{d}(S_{0})$, for any point $S_{0}$ in the region. The Kriging method takes full account of the correlation of PM_2.5_ site data by calculating the variance function of the sample.

### 3.4. LUR-GBM Model

Figure 2 shows the research framework. A total of six models (BPNN, DNN, RF, XGBoost, LightGBM, LUR-GBM) were developed in this study. Given that the shortcomings of machine learning models in selecting appropriate predictor variables can be addressed by applying LUR, this study aims to use an integrated approach combining LUR and machine learning models to improve the estimation of regional PM_2.5_ daily concentrations in China for the period 2016 to 2021. First, a traditional LUR is used to identify significant predictor variables. A deep neural network, random forest and XGBoost algorithms were then used to fit a predictive model based on the variables selected by the LUR model. Data partitioning, 10-fold cross-validation, external data validation and seasonal and year-based validation methods were used to validate the robustness of the developed models. Specifically, the significant predictor variables identified through the stepwise variable selection of the LUR procedure were applied to LightGBM to improve the accuracy of PM_2.5_ change predictions. A hybrid spatial prediction model combining the strengths of LUR in identifying the most influential emission projections with the predictability of machine learning in estimating non-linear trends will be more effective than techniques that rely on LUR or machine learning alone. In order to fully consider the problem of spatial correlation of monitoring station data in PM_2.5_ mass concentration estimation and to improve the accuracy of PM_2.5_ spatial estimation, this paper introduces the Kriging method and constructs a spatio-temporal LUR-GBM model, which provides a new idea to solve the complex spatial relationship in PM_2.5_ estimation. The LUR-GBM model takes into account the influence of the PM_2.5_ value at any point in space on the values of other stations in the surrounding neighbourhood, using the spatial location estimate $R_{d}$ calculated by the Kriging method and the temporal location $R_{t}$ of that point as input variables to the model. The LUR-GBM model can be expressed as Equation (8).
(8)$${\mathrm{EPM}}_{2.5}=Model(\;R_{d},R_{t},cro,for,ind,altitude,AOD,TEM,WS,LAT,LON)$$
where ${\mathrm{EPM}}_{2.5}$ is the LUR-GBM model PM_2.5_ estimate, Model is the LUR-GBM model, LAT is the latitude and LON is the longitude.

### 3.5. Accuracy Evaluation

To fully evaluate the performance of the LUR-GBM model, a ten-fold cross-validation (10-CV) based on samples, sites and time was used, and the computed results were compared with BPNN, DNN, RF, XGBoost and LightGBM. Three indicators, coefficient of determination (R^2^), root mean square error (RMSE), mean prediction error (MAE) and mean absolute percentage error (MAPE), were calculated separately from the model prediction results to test the model performance [42]. R^2^ is a measure of the degree of linear correlation between variables and reflects the proportion of the variation in the dependent variable that can be explained by the independent variable. Therefore, the coefficient of determination was selected as one of the indicators for model evaluation in this study [43]. Each evaluation indicator is calculated using the following formula:(9)$$RMSE=\sqrt{\frac{\sum (PM_{F}-PM_{T}{)}^{2}}{N}}$$
(10)$$R^{2}=\frac{COV(PM_{F}-PM_{T})}{\sqrt{v\mathrm{ar}[PM_{F}]v\mathrm{ar}[PM_{T}]}}$$
(11)$$MAE=\frac{\sum (PM_{F}-PM_{T}{)}^{2}}{N}$$
(12)$$MAPE=\frac{\sum \left|PM_{F}-PM_{T}\right|}{N}\times \frac{100\%}{PM_{T}}$$
where $PM_{F}$ is the predicted PM_2.5_ value; $PM_{T}$ is the measured PM_2.5_ value; N is the number of samples.

## 4. Results and Analysis

### 4.1. Correlation Analysis of PM_2.5_ Concentrations and Impact Factors

The results of the bivariate correlation analysis between PM_2.5_ concentration and influencing factors are shown in Table 2. Within the land use sub-categories, arable land, forest land, grassland and urban and rural industrial and mining residential land all have a strong influence on the change of PM_2.5_ concentration. Among the road traffic data, highways and major arterial roads had a strong influence on the change of PM_2.5_ concentration, while topography, forest land, grassland and unused land had a negative relationship with PM_2.5_ concentration, and road traffic, urban and rural industrial and mining residential land maintained a positive relationship with PM_2.5_ concentration. PM_2.5_ concentrations are negatively correlated with factors such as woodland, grassland, water, altitude, precipitation and relative humidity. PM_2.5_ concentrations are positively correlated with factors such as industrial and mining settlements, barometric pressure, temperature, population density and road length. *p*-values represent the level of significance. *p*-values are highly significant at α = 0.01 for correlation and at α = 0.05 for correlation. Table 2 shows that population was significantly correlated at α = 0.05, and all other modelling variables were highly significant at the α = 0. 01 level, all passing the variable significance test.

### 4.2. Model Performance

Using China as the study area, data from 1 January 2016 to 31 December 2021 were selected, and the training dataset and the test validation dataset were selected by multiple random sampling. The training set was 70%, the test validation set was 30% and the experimental evaluation was repeated and averaged as the evaluation result of the model. Training of the LUR-GBM model was completed via Python 3.7. The LUR-GBM model was trained using the target factors selected by bivariate correlation analysis as features of the model and the PM_2.5_ concentrations at the monitoring stations as supervised values. The LightGBM model had the following detailed parameters: *Base learner = GBDT*, the number of base learners is 100, *Num_leaves = 31*, *Learning_rate = 0.05*, *Feature_fraction = 0.9*, *Bagging_fraction = 0.8*, *Bagging_freq = 5.*

Table 3 shows the performance of the machine learning models, with R^2^ ranging from 0.76 to 0.98 for the five machine learning models in a sample-based cross-validation. The R^2^ of both the LightGBM and LUR-GBM models considering site data correlation, was greater than 0.9, with the LUR-GBM model performing best. The RMSE of the models ranged from 6.43 to 11.37 μg/m^3^, with the LUR-GBM model having the lowest RMSE value and the BPNN model having the highest RMSE value (11.37 μg/m^3^). The MAE was 4.17 to 8.35 μg/m^3^, with the LUR-GBM model having the lowest MAE value of 4.17 μg/m^3^, followed by the LightGBM model at 4.56 μg/m^3^.

In the site-based cross-validation, the R^2^ values of the LightGBM and LUR-GBM models considering geographical correlation and temporal variation were significantly higher than those of traditional machine learning models such as XGBoost, RF, and BPNN, but the R^2^ values were lower compared to those of the sample-based cross-validation because of the significant spatial heterogeneity of PM_2.5_ distribution in space. The LUR-GBM model has the highest R^2^ value of 0.91, followed by the LightGBM model, and the BPNN model performed the worst. Comparing the RMSE and MAE metrics, the RMSE and MAE values of the LightGBM and LUR-GBM models were significantly lower than those of other traditional machine learning models, with the LUR-GBM model performing best with RMSE and MAE values of 7.46 μg/m^3^ and 5.01 μg/m^3^, respectively.

The LUR-GBM model performs well at the spatial scale, taking full account of the relevance of site data. The relatively poor performance of the time-based cross-validation models is due to the fact that the PM_2.5_ distribution varies significantly in time scale. The R^2^ of each machine learning model ranged from 0.56 to 0.89, with the LUR-GBM model performing the best with an R^2^ value of 0.89, followed by the LightGBM model and the BPNN model performing the worst. Comparing the RMSE and MAE indices, the LUR-GBM model had the lowest RMSE and MAE values of 7.07 μg/m^3^ and 4.95 μg/m^3^, respectively, while the BPNN model had the highest RMSE and MAE values of 13.28 μg/m^3^ and 9.69 μg/m^3^. This indicates that the LUR-GBM model, which takes into account time variation, performs better on the time scale.

Figure 3 shows the scatter plot of the PM_2.5_ concentrations estimated by the BPNN, RF, DNN, XGBoost, LightGBM and LUR-GBM models fitted to the PM_2.5_ concentrations measured at the ground monitoring sites. As can be seen from Figure 3, the LightGBM model and LUR-GBM model outperform traditional machine learning models such as BPNN, DNN, RF and XGBoost. The reason for this is that the LightGBM model and the LUR-GBM model take into account site data and temporal variation and can better characterise the spatial and temporal characteristics of PM_2.5_. The scatter density plots drawn by the LightGBM model and the LUR-GBM have a fit ratio R^2^ of 0.91 and 0.98, respectively, indicating that the LUR-GBM model is the best fit. The LUR-GBM is based on the LightGBM model with the introduction of the Kriging method, which improves the accuracy of PM_2.5_ estimation by calculating the variance function and taking full account of the spatial correlation of station data. The BPNN estimated ground-level PM_2.5_ mass concentrations were the least well fitted, grossly underestimating PM_2.5_ values and performing the worst. The overall error values of our model are small, but as some extreme phenomena can occur, such as dust storms in places like Xinjiang, most of the areas where the detection values exceed 200 µg/m^3^ are in these areas. This leads to situations where some of the predicted data can deviate significantly from the true values, which, combined with the fact that the monitoring stations in this part of the country are not fully covered and the large distances between the various monitoring stations, leads to large deviations. Furthermore, the model is based on daily regional PM_2.5_ mass concentration data for 2016–2021 in China, taking into account regional variability and, therefore, a small number of deviations in the predicted values. We can find by the value of MAPE that the average error of BPNN is more than 30% at maximum, and the value of MAPE of the LightGBM model and LUR-GBM model among the six models is less than 20%, where the average error of LUR-GBM model is 15.304%. A comprehensive comparison of the six machine learning models showed that the LUR-GBM model had the best prediction performance, followed by the LightGBM model, while the BPNN had the worst prediction performance among the six models.

We validated the six models using annual average data from 2016 to 2021, and Figure 4 shows the scatter density plots of PM_2.5_ concentrations estimated by the BPNN, RF, DNN, XGBoost, LightGBM and LUR-GBM models fitted to the actual PM_2.5_ concentrations measured at ground monitoring stations. The overall performance of the six models was better than the performance of the predictions of daily concentrations. This is due to the fact that annual concentrations are less variable and volatile and that annual values are less affected by extreme values. As can be seen from Figure 4, the LightGBM and LUR-GBM models outperformed traditional machine learning models such as BPNN, DNN, RF and XGBoost, with R^2^ values of 0.82 and 0.866, respectively, in terms of goodness of fit. BP and DNN had the worst fit performance of 0.75 and 0.79, respectively. The best RMSE values among the six models were 5.571 ug/m^3^ for the LightGBM model and 5.291 ug/m^3^ for the LUR-GBM model, while the worst was 6.669 ug/m^3^ for the BP. In terms of MAE values, the LUR-GBM model had a minimum of 4.021 ug/m^3^ and the BP had a maximum of 6.669 ug/m^3^. In terms of MAPE values, all six models were less than 15%, with the LUR-GBM model being the smallest at 10.71%. A comprehensive comparison of the six machine learning models shows that the LUR-GBM model had the best prediction performance, followed by the LightGBM model, while the BPNN has the worst prediction performance among the six models. The values of RMSE, MAE and MAPE all decreased compared to the annual concentration data. However, the R^2^ was considerably lower compared to the annual concentration data, mainly because the annual concentration data was too small compared to the daily concentration data for a good fit.

Figure 5 shows a scatter density plot of the PM_2.5_ concentrations estimated by the LUR-GBM model on a seasonal scale and fitted to the PM_2.5_ concentrations measured at ground-based monitoring stations. A ten-fold cross-validation based on samples showed that R^2^ (0.98) was highest in autumn. The highest RMSE (12.54 μg/m^3^) in spring and MAE (7.61 μg/m^3^) in winter were the seasons where the higher correlation between surface temperature and PM_2.5_ contributed to the difference in R^2^. In contrast, the lowest R^2^ (0.91) and the lowest RMSE (4.34 μg/m^3^) and MAE (3.01 μg/m^3^) were recorded in summer. The lower estimation error in summer was due to the lower ground level PM_2.5_ mass concentration due to frequent rainfall and the higher estimation accuracy. Overall, the LUR-GBM model performed well on seasonal scales and was able to predict the distribution of PM_2.5_ mass concentrations on seasonal scales. To test the accuracy of the LUR-GBM model simulation, a linear correlation analysis was performed between the simulated PM_2.5_ model values at the validation site and the actual measured values at the site. As shown in Figure 6, the fitted R^2^ for 2016, 2017, 2018, 2019, 2020 and 2021 are 0.98, 0.97, 0.97, 0.98, 0.98 and 0.98 respectively. The fitted R^2^ for the 6-year mean was 0.98, and the accuracy of the annual mean inversion was slightly higher than the accuracy of the quarterly mean inversion. The results show that the inversion of PM_2.5_ concentrations by constructing the LUR-GBM model is highly accurate. The training and validation results of the LUR-GBM model for 2016, 2017, 2018, 2019, 2020 and 2021 data showed a mean RMSE value of 9.29 μg/m^3^ and a mean MAE value of 5.819 μg/m^3^.

### 4.3. Spatial and Temporal Distribution Characteristics of PM_2.5_ Mass Concentration in China

#### 4.3.1. Seasonal Distribution Characteristics

The seasons were first divided according to the climatic conditions of the Chinese region as a whole: spring from March to May, summer from June to August, autumn from September to November and winter from December to February. The spatial distribution of seasonal average PM_2.5_ concentrations is shown in Figure 7, mostly high in winter and low in summer, falling in spring and rising in autumn. Summer air quality is good in all cities, with pollution below 35 μg/m^3^ in most areas. East China, Central China and the Fenwei Plain are the most polluted in winter, with most cities in the region exceeding 70 μg/m^3^. Concentrations are higher in the north than in the south in spring and more serious in autumn, mainly in East China and Xinjiang. The very highest values of seasonal pollution occur in winter in Xinjiang, reaching above 100 μg/m^3^. Apart from the relatively good air quality in summer, Xinjiang has a certain degree of pollution in all other seasons, but it is still among the most polluted of all cities in the country in summer, and the pollution is at high levels throughout the region in winter. The overall air quality in southern China is good, with little difference between spring, summer and autumn, based on less than 40 μg/m^3^, and relatively serious pollution in winter, mostly concentrated in Hunan and Jiangxi provinces.

#### 4.3.2. Spatial and Temporal Distribution of PM_2.5_ Concentrations in China

Figure 8 shows the spatial distribution of annual average PM_2.5_ mass concentrations in the Chinese regions as estimated by the LUR-GBM model. The PM_2.5_ values estimated by the LUR-GBM model are consistent with the distribution trend of the measured values at ground monitoring stations, with an annual average PM_2.5_ mass concentration of 38 μg/m^3^ from 2016 to 2021. The average PM_2.5_ concentrations from 2016 to 2021 show a spatially higher level in the north than in the south. In 2016, the annual average PM_2.5_ concentration was 47 μg/m^3^, with heavy pollution mainly concentrated in southern Hebei, Shandong Province (except Weihai and Yantai) and the borders of Shandong, Anhui and Jiangsu provinces, with some cities reaching severe pollution levels. Moderate pollution is mainly concentrated in the northeast and north of the Yangtze River, with some cities south of the Yangtze being lightly polluted and some meeting air quality standards. In 2017, the overall national pollution situation improved somewhat, with an annual average concentration of 43 μg/m^3^. Pollution concentrations in the southern part of the region were not significantly different from 2016, with fewer areas of severe pollution in Shandong Province and some cities having reduced from heavy to moderate pollution, although pollution in some cities in Anhui increased; some areas south of the Yangtze River met air quality standards with annual average concentrations below 35 μg/m^3^. In 2018, the annual average urban PM_2.5_ concentration in China was 39 μg/m^3^, with significant improvement in air quality conditions across the region. Heavy pollution areas are concentrated in the central region provinces such as southern Gansu, southern Shaanxi, Shanxi, Henan and northern Hubei. North of the Yangtze River has been reduced from moderate pollution to light pollution, and the number of areas south of the Yangtze River air quality standards increased significantly. In 2019, the national average PM_2.5_ concentration was 36 μg/m^3^, 57 μg/m^3^ in Beijing, Tianjin, Hebei and surrounding areas, and 41 μg/m^3^ in the Yangtze River Delta, a decrease of 2.4% from 2018. The concentration of PM_2.5_ in the Fenwei Plain was 55 μg/m^3^, with serious pollution concentrated in the Fenwei Plain, Eastern China and Xinjiang. In 2020, the national PM_2.5_ concentration was 33 μg/m^3^. The overall trend of contamination was down due to the impact of the epidemic. In Beijing, Tianjin, Hebei and surrounding areas, including key areas such as the Fenwei Plain, emissions of air pollutants remain high, and PM_2.5_ concentrations remain high. The ratio of good days in prefecture-level cities and above was 87.5% in 2021, an increase of 0.5 percentage points year-on-year; the PM_2.5_ concentration was 30 μg/m^3^. With the exception of Xinjiang and some provinces in East and Central China, there were fewer areas of overall pollution. Polluted provinces such as Henan, southern Hebei and northern Shaanxi all saw varying degrees of decline. China’s series of environmental protection measures in recent years have been effective in reducing the concentration of PM_2.5_ pollution, and the government should continue to maintain its environmental monitoring efforts to improve air quality standards. The above analysis shows the spatial distribution of annual average PM_2.5_ concentrations in Chinese cities from a global perspective.

Spatially, as a whole, the polluted regions show more serious pollution in the east than in the west, which is consistent with China’s overall economic development and urbanisation and population distribution. Pollution is serious in northern China, with pollutants concentrated in southern Hebei, northern Henan and western Shandong, with average concentrations above 70 μg/m^3^, due to dense industry and serious pollutant emissions in northern China. Central China and the Sichuan Basin also have greater air pollution due to the economically developed and densely populated central China, where intense human activity has led to increased pollutant emissions, and the special topography of the Sichuan Basin, which is not conducive to the dispersion of pollutants. Due to its southerly location, coastal position, high rainfall and low air pollution, the average PM_2.5_ concentration in southern China is below 30 μg/m^3^, which is lower than the national average annual concentration. In addition, the Xinjiang region also experienced more serious air pollution due to the frequent dust storms and poor air quality in the Taklamakan Desert in Xinjiang.

### 4.4. Fitting Assessment of PM_2.5_ Concentrations in Typical Chinese Cities

The Beijing-Tianjin-Hebei Urban Agglomeration, the Yangtze River Delta and the Fenwei Plain are areas with high emission intensity per unit area of air pollution sources in China, and these three regions are also key areas identified by the state for air pollution prevention and control [44]. We fitted PM_2.5_ concentrations to 10 typical cities in heavily polluted areas, which are cities with large populations in Beijing, Tianjin and Hebei, the Fenwei Plain and the Yangtze River Delta and have a relatively large number of observation sites. As shown in Figure 9, the R^2^ of the fit was above 98% for all 10 cities, with Hangzhou and Hefei having the highest accuracy in terms of RMSE and MAE values and Shijiazhuang and Tianjin having poorer results. We found that the accuracy of northern cities was lower than that of southern cities, and the main reason for this is that northern cities such as Beijing and Tianjin are affected by sandstorms, while the lower winter temperatures and more snow and ice lead to more complex aerosol types, which affects the accuracy of the model.

## 5. Discussion

(1)To verify that the PM_2.5_ concentration prediction based on the LUR-GBM model was more accurate, validation was carried out from the perspective of different datasets and different control models. In terms of cross-sectional datasets, by predicting PM_2.5_ concentrations based on sample-based datasets, site-based datasets and time-based datasets, the LUR-GBM model was found to have the highest prediction accuracy with sample-based datasets. In particular, compared to the PM_2.5_ concentration prediction based on the station dataset, the result of the sample dataset-based prediction improved R^2^ by 7.69%, reduced RMSE by 13.81% and reduced MAE by 16.77%. Compared to the PM_2.5_ concentration prediction based on the time dataset, the result of the sample dataset-based prediction improved R^2^ by 10.11%, reduced RMSE by 9.05% and reduced MAE by 15.76%. From the models, the LUR-GBM model had improved prediction accuracy over BPNN, DNN, RF, XGBoost and LightGBM. Compared to the BPNN model, the LUR-GBM model improved R^2^ by 42.63%, reduced RMSE by 41.15% and reduced MAE by 48.45% on average. Compared to the DNN model, the LUR-GBM model improved R^2^ by an average of 16.31%, reduced RMSE by 34.23% and reduced MAE by 42.37%. Compared to the RF model, the LUR-GBM model improved R^2^ by 12.99%, reduced RMSE by 33.22% and reduced MAE by 34.20%. Compared to the XGBoost model, the LUR-GBM model improved R^2^ by an average of 10.29%, reduced RMSE by 23.31% and reduced MAE by 22.54%. Compared to the LightGBM model, the LUR-GBM model improved R^2^ by an average of 7.33%, reduced RMSE by 7.46% and reduced MAE by 10.47%.(2)The distribution of PM_2.5_ concentrations in China is characterised by high winter and low summer, falling in spring and rising in autumn. In winter, PM_2.5_ pollution is most severe in areas such as the Fenwei Plain. In spring, PM_2.5_ concentrations are higher in northern China than in southern regions. In autumn, PM_2.5_ pollution is most severe in eastern China and Xinjiang. In summer, air quality is better throughout the country, except in Xinjiang.(3)A further decrease was found in the national average PM_2.5_ concentration from 47 ug/m^3^ to 30 ug/m^3^ from 2016 to 2021. Seriously polluted areas are concentrated in the Fenwei Plain, Eastern China and Western Xinjiang. In terms of the spatial distribution of PM_2.5_ concentrations, China’s pollution regions as a whole are characterised by higher levels in the east than in the west. North China is the most polluted region, mainly including southern Hebei, northern Henan and western Shandong. This was followed by greater air pollution in Central China, the Sichuan Basin and Xinjiang. Southern China has the lowest PM_2.5_ concentration and the best air quality.(4)PM_2.5_ concentration predictions for ten typical cities in heavily polluted regions of China were studied and discussed and found to be less accurate in northern cities than in southern cities. Hangzhou and Hefei had the highest forecast accuracy, while Shijiazhuang and Tianjin had a lower forecast accuracy.

## 6. Conclusions

In this paper, a typical hybrid model LUR-GBM is proposed based on the PM_2.5_ observation data of China from 2016 to 2021. The spatial and temporal distribution of PM_2.5_ concentrations was estimated using AOD data from satellite remote sensing inversions as well as conventional meteorological observation elements, land use and road data. By analysing the spatial and temporal patterns of PM_2.5_ and its influencing factors, this paper clarifies the changes in PM_2.5_ at different time scales and the underlying mechanisms in recent years and summarises the general patterns of PM_2.5_ concentrations in the spatial and temporal distribution in China. Therefore, the inversion of PM_2.5_ can help to grasp the regional variation process of PM_2.5_ in time and space by taking into account the land use information, correlation and spatio-temporal heterogeneity. This study provides a scientific basis for the prevention and control of regional PM_2.5_ pollution and a new way of thinking for management departments to obtain data on the spatial distribution of PM_2.5_ concentrations. The LUR-GBM method is a better solution to the problem of spatial heterogeneity of research objects.

The recommendations in this paper are as follows:(1)Improve joint prevention and control mechanisms in different regions. The formation and sources of PM_2.5_ are complex, and it is difficult to control a single source and a single city to radically reduce the pollution. Analysis of the spatial distribution of PM_2.5_ on a regional scale can further provide reliable information to support the establishment of improved regional joint prevention and control mechanisms in order to better address urban air pollution.(2)Fine-grained regulation of pollution levels by zoning. Pollution prevention and control measures are formulated according to the different geographical features, meteorological conditions and economic development of different regions, taking into account local conditions. Differential control management for heavily polluted areas and general areas. The relevant government departments should speed up the improvement of early warning and treatment of heavily polluted areas.(3)Implementation of seasonal differentiation of control. This study found significant differences in PM_2.5_ concentrations between seasons, requiring the implementation of targeted prevention and control measures. Measures such as reducing pollution through artificial precipitation, imposing restrictions on motor vehicles and reasonable heating.(4)Strengthen the control of pollution at the source. There is a need to increase energy restructuring and energy conservation and emission reduction efforts to prevent and control air pollution at the source. Rational allocation of functional tasks of agency staff to areas with different PM_2.5_ levels through predictive warning. Timely release of information on pollution sources to achieve the transformation from governance to prevention.

## Figures and Tables

**Figure 1 ijerph-19-06292-f001:**
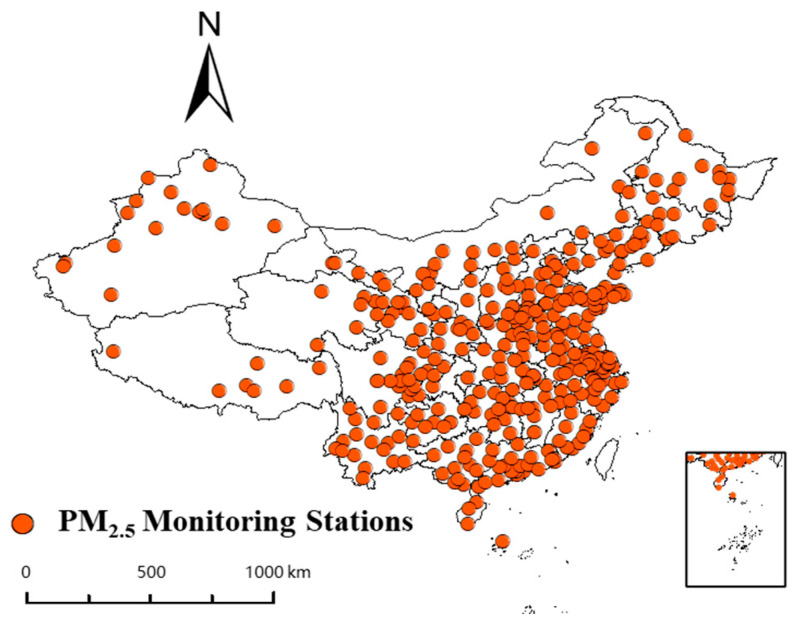
Distribution of PM_2.5_ ground monitoring stations in China.

**Figure 2 ijerph-19-06292-f002:**
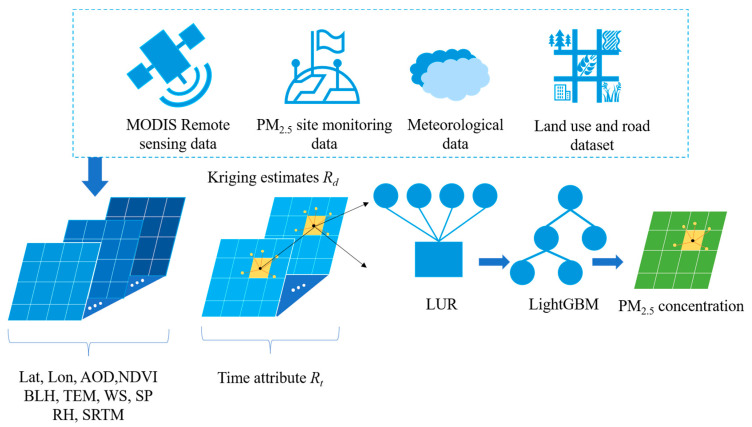
LUR-GBM model structure diagram.

**Figure 3 ijerph-19-06292-f003:**
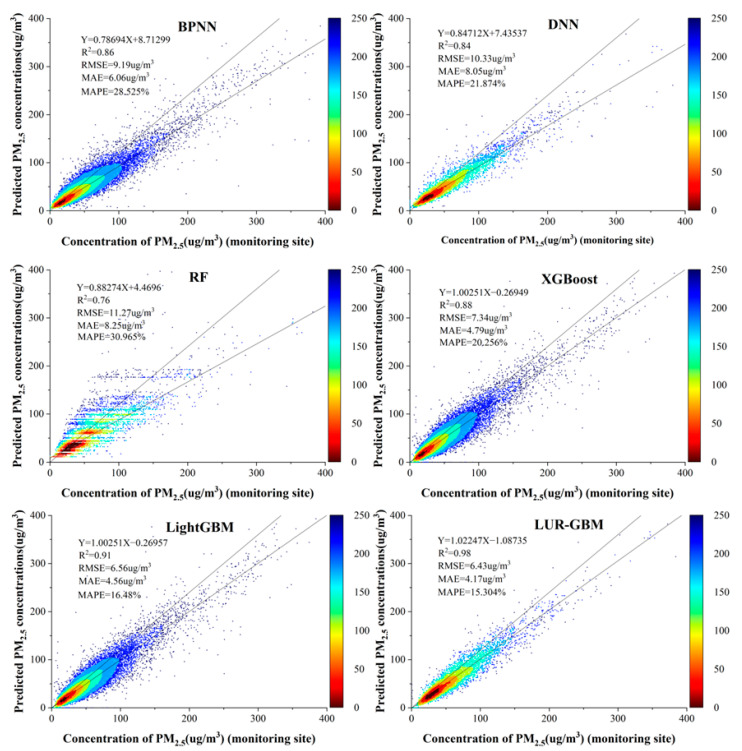
Six-model scatter point density map.

**Figure 4 ijerph-19-06292-f004:**
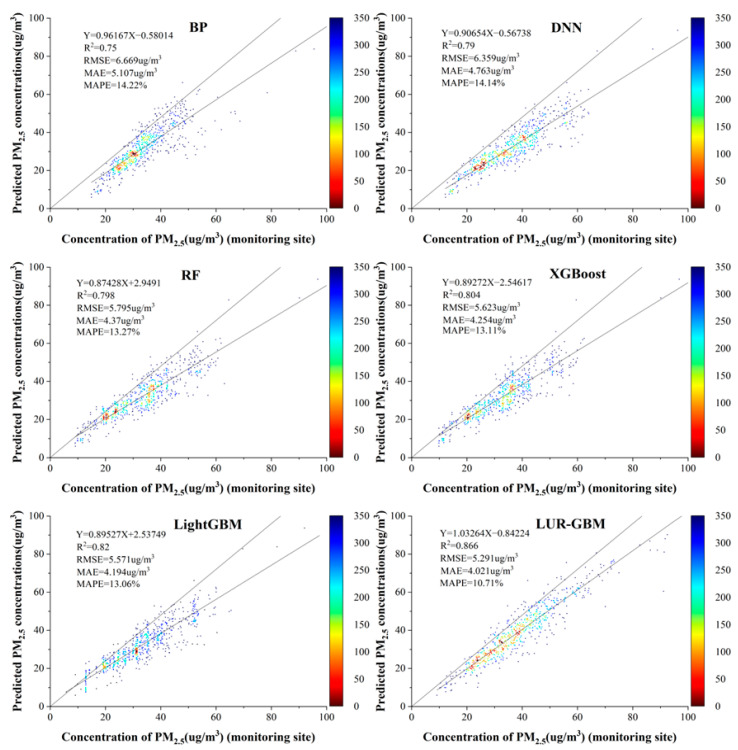
Scatter density plots of annual mean concentrations for the six models.

**Figure 5 ijerph-19-06292-f005:**
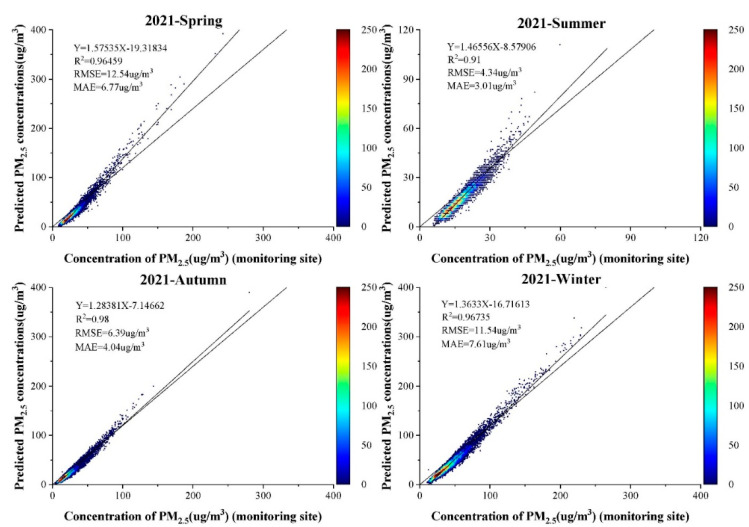
Scatter density map based on LUR-GBM model for all seasons in 2021.

**Figure 6 ijerph-19-06292-f006:**
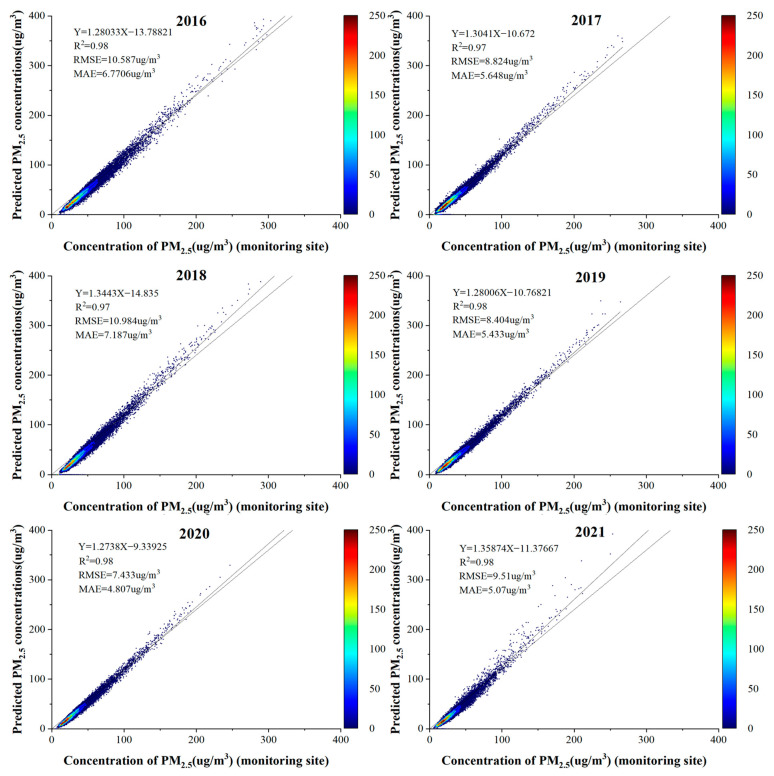
PM_2.5_ concentration simulation 2016–2021 scatter density map.

**Figure 7 ijerph-19-06292-f007:**
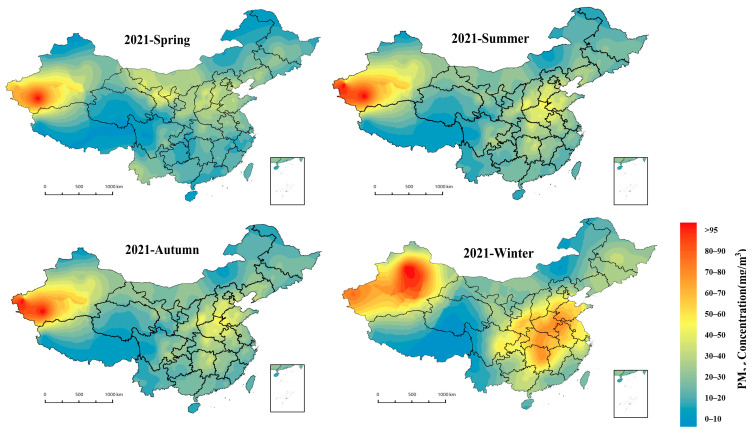
Spatial distribution of quarterly inversions based on the LUR-GBM model for 2021.

**Figure 8 ijerph-19-06292-f008:**
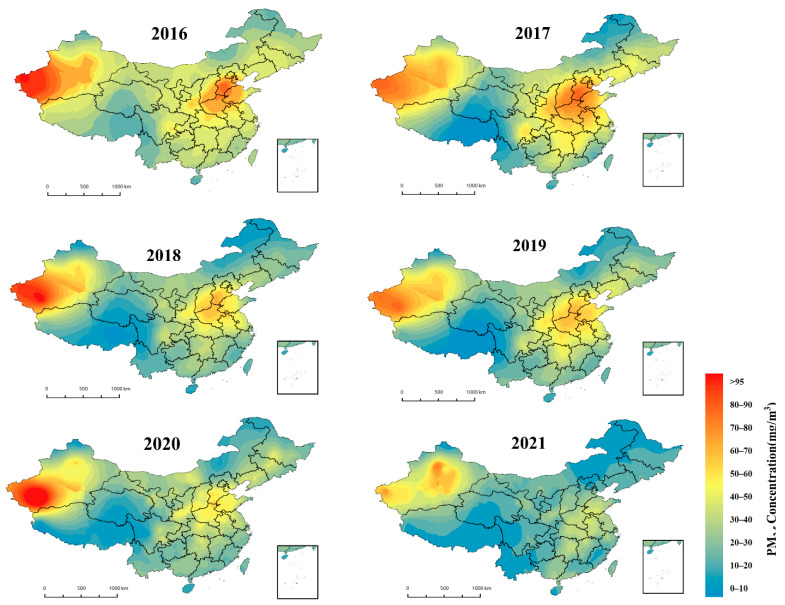
Simulated annual average distribution of PM_2.5_ concentrations based on the LUR-GBM model.

**Figure 9 ijerph-19-06292-f009:**
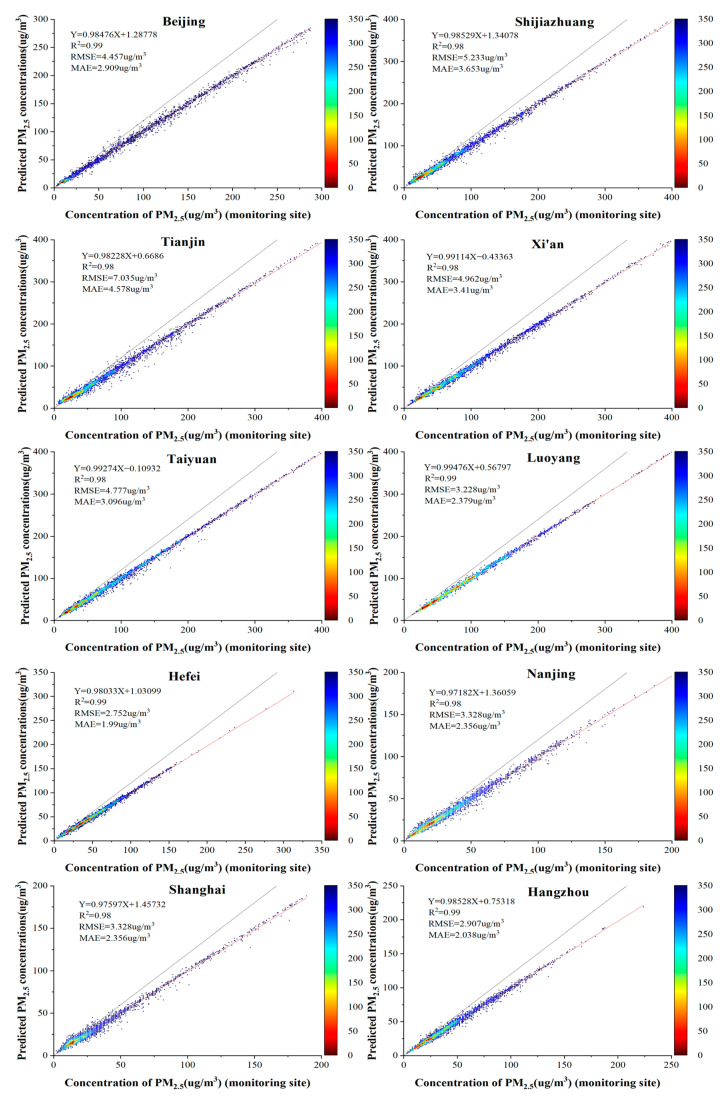
Scatter plot of model results for the 10 cities in the heavily polluted areas based on the LUR-GBM model.

**Table 1 ijerph-19-06292-t001:** Classification and description of independent variables.

Variable Type	Variable Name	Unit	Variable Description
Land type	cro	%	Cropland
	for	%	Forest
	gra	%	Grass
	wat	%	Water
	ind	%	Industrial and residential
	sem	%	Seminatural
Terrain and landforms	altitude	m	Altitude
Population	pop	people	Population
Road traffic	hig	m	Highway
	maj	m	Major road
	hm	m	Sum of highway and Major road
	min	m	Minor road
Meteorological elements	GST	°C	0 cm Surface temperature
	SSD	h	Sunshine hours
	PRS	hPa	Pressure
	TEM	°C	Temperature
	RHU	%	Relative humidity
	PRE	mm	Precipitation
	WIN	m/s	Wind speed

**Table 2 ijerph-19-06292-t002:** Results of bivariate correlation analysis between PM_2.5_ concentration and impact factors.

Independent Variable	Pearson Correlation	*p*	Independent Variable	Pearson Correlation	*p*
cro	0.343	0.003	pop	0.310	0.021
wat	−0.059	0.002	altitude	−0.559	0.000
for	−0.379	0.000	GST	0.178	0.000
gra	−0.299	0.000	SSD	0.018	0.000
ind	0.322	0.000	PRS	0.302	0.000
sem	−0.134	0.000	TEM	0.523	0.000
hig	−0.084	0.000	RHU	−0.215	0.001
maj	0.187	0.002	PRE	−0.346	0.004
hm	0.177	0.000	WIN	0.415	0.000
min	0.125	0.002			

**Table 3 ijerph-19-06292-t003:** Comparison of results of various models.

	Based on Samples			Based on Sites			Based on Time		
	R^2^	RMSE	MAE	R^2^	RMSE	MAE	R^2^	RMSE	MAE
BPNN	0.76	11.27	8.35	0.65	11.26	9.34	0.56	13.28	9.69
DNN	0.84	10.33	8.05	0.78	11.09	8.86	0.77	10.43	7.67
RF	0.86	9.19	6.08	0.81	11.03	7.46	0.79	11.27	8.03
XGBoost	0.88	7.34	4.79	0.83	10.54	6.78	0.81	9.86	6.93
LightGBM	0.91	6.56	4.56	0.85	8.32	5.76	0.83	7.86	5.49
LUR-GBM	0.98	6.43	4.17	0.91	7.46	5.01	0.89	7.07	4.95

## Data Availability

MODIS remote sensing data from the NASA Goddard Space Flight Center website (https://ladsweb.modaps.eosdis.nasa.gov/ (accessed on 20 April 2022)). PM_2.5_ site monitoring data from the national real-time urban air quality release platform (http://106.37.208.233:20035/ (accessed on 20 April 2022)). Meteorological data from the European Centre for Medium-Range Weather Forecasts website (https://www.ecmwf.int/ (accessed on 20 April 2022)). Land use data from the China Geographic Monitoring Cloud Platform (http://www.dsac.cn/ (accessed on 20 April 2022)).

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
