# Peer review of "Spatio-Temporal Characteristics of PM2.5 Concentrations in China Based on Multiple Sources of Data and LUR-GBM during 2016–2021"

_ijerph, 2022, doi:10.3390/ijerph19106292_

Round 1
Reviewer 1 Report
This paper seeks to analyze the spatial and temporal characteristics of PM2.5 Concentrations in China Based on Multiple Sources of Data and LUR-GBM during 2016–2021. The paper is well written in terms of writing style and the English language.
MINOR REVIEW
I suggest the first word in the title be rephrased to read "Spatio-Temporal" instead of using "Spatial". I know the author wanted to vary the commonly used 'spatio'.
What is the reason for the choice of the period (2016-2021)?
Why is 'table 2' discussed under Data and Methods??
The author has 'Data and Methods' as Section 2, but introduces section 3 also as Methods. The author should either merge section 2 and 3 or separate them completely.
The equations beneath the modeling process should be well separated not to look like one equation.
line21: Introduce a hyphen in LURGBM for consistency. Also on line 291
line23: Write the full meaning of BP
line24: Write the full meaning of XGBoost
line63 and 64: Statements should be referenced
line 80: Full meaning of NO2 should be stated although it's understood generally
line140-141: Is the last part of point 2 a conclusion or discussion? That shouldn't be under the introduction if it is your own results.
line 314: What is the meaning of PM subscript 's'. This wasn't stated.
Author Response
We are thankful for your suggestions and comments. Your comments have greatly improved the quality of our articles. Thank you for taking the time to review the manuscript. We have also reworked the figures, results, discussion section and also revised the other part of the manuscript.

Reviewer 2 Report
The problem to be solved is interesting and worth to be published. However, some improvements are needed . Here are some comments to improve the paper (from my point of view).
- The significant problem is the presentation of the paper. The presentation needs to emphasize the problem to be solved. There is writing that can be tightened up to eliminate repetition and points that are not central to the results. There are too many repetitions.
- The main problem is the lack of clear description of the motivation and significant discussions related to the presented results. More explanations and discussions should be addressed related to the results and method.
- Table 2 provides the P-values, however there are no discussions about them. Please present the corresponding discussions of P-values in Table 2.
- From statistical point of view, at significant level=0.05 there is no Significant Effect for any of the mentioned factors on the change of PM concentration. Can you please explain such interesting observation?
- I think the proposed model in equation (12) can be reduced based on the P-values in Table 2. That is this equation can be given using only the factors that have significant effect instead of all the factors.
- I think figure 2 can be deleted, it is a well-known process.
- Equations (2) and (3) can be merged.
- Line 232, please change "x_1,x_2,x_n" to "x_1,x_2,...,x_n"
- Equations (5) and (6) can be merged.
- Equations (7) and (8) can be deleted. You can just mention that, the estimator is a minimum variance unbiased estimator.
- Equations (10) and (11) can be merged.
- As you mentioned, the LUR-GBM model is expressed by equation (12) that is given based on 23 factors (input variables). However, the equations in Section 4 are given based on only one factor X. What is X? More discussions are needed.
- The R^2, RMSE and MAE in equations (13)-(15) are given based on PM_F and PM_T that are the inverse and measured PM value, respectively. What do you mean by the inverse PM value? Please explain.
- Please check the title of Y axis of Figures 5 and 6.
- From Figure 9, it is obvious that there are no significant differences among the 10 cities. Therefore, no need to study each city individually.
- The paper is too long in view of the new contribution.
Author Response

(The authors gave the same response as above.)

Reviewer 3 Report
The study is interesting, and the methodology used relatively results novel.
Minor comments:
The last paragraph does not correspond to the development executed in the work in the introduction section. For example, the phrase 'The second section focuses on the data 146 sources and methods used in this study' is not adequate because you introduce correlation results in this section.
Figure 2. Although it is interesting, it is general; the authors should update it to the study's variables to facilitate its applicability to readers.
Equations 1 and 4. It must be applied to the study variables, not in general.
Fig 4 and 5. The regression line must be the same color.
Major comments:
LUR-GBM model used daily PM2.5 concentrations. It would be interesting if the authors could test the model set for annual data since the air quality objectives are given as annual averages.
Equations: They must be sustained by references.
The RMSE and MAE values are absolute; it would be relevant to put them in relative values since figure 4 shows an inaccuracy > 20%. The model fits more for values < 150-200 µg/m3.
Author Response

(The authors gave the same response as above.)

Round 2
Reviewer 2 Report
I think the revised version is a good answer to existing questions, and is suitable for publication in the journal.
Author Response
We have made improvement on the result. Thank you for your comments. Have a nice day!

Reviewer 3 Report
Dear authors,
Allow me thank your the valuable answers provided to the reviewer's comments. The manuscript has notably improved. Nevertheless, some enhancement are yet needed.
Comments to the document titled 'Response to Reviewer 3 Comments'
Point 1: Table 2 is not adequate in Source data section (line 159). Its discussion appears in line 277.
Point 2: It is ok.
Point 3: It is ok.
Point 4: It is ok.
Point 5: A summary concerning the explication of point 5 should be included in the paper. This point of view will be interest for potential readers.
Point 6: The explication provided in the point 6 should be explained within the manuscript.
Best regards,
Author Response
We are thankful for your suggestions and comments. Thank you again for taking the time to help review it. We have changed the results section. Have a good day!

Round 3
Reviewer 3 Report
Dear authors,
Thank you for the effort carried out for providing answers to the reviewer's comments.
Best regards,